mechanics/mathematical physics

Herglotz type variational problem, Birkhoffian system, Noether theorem, delta derivative, time-scale

**Author for correspondence:**
Yi Zhang
e-mail: zhy@mail.usts.edu.cn

# Time-scales Herglotz type Noether theorem for delta derivatives of Birkhoffian systems

## Xue Tian[1] and Yi Zhang[2]

[1]School of Science, Nanjing University of Science and Technology, Nanjing 210094, People's Republic of China
[2]College of Civil Engineering, Suzhou University of Science and Technology, Suzhou 215011, People's Republic of China

(iD) YZ, 0000-0002-7703-1185

The time-scales theory provides a powerful theoretical tool for studying differential and difference equations simultaneously. With regard to Herglotz type variational principle, this generalized variational principle can deal with non-conservative or dissipative problems. Combining the two tools, this paper aims to study time-scales Herglotz type Noether theorem for delta derivatives of Birkhoffian systems. We introduce the time-scales Herglotz type variational problem of Birkhoffian systems firstly and give the form of time-scales Pfaff–Herglotz action for delta derivatives. Then, time-scales Herglotz type Birkhoff's equations for delta derivatives are derived by calculating the variation of the action. Furthermore, time-scales Herglotz type Noether symmetry for delta derivatives of Birkhoffian systems are defined. According to this definition, time-scales Herglotz type Noether identity and Noether theorem for delta derivatives of Birkhoffian systems are proposed and proved, which can become the ones for delta derivatives of Hamiltonian systems or Lagrangian systems in some special cases. Therefore, it is shown that the results of Birkhoffian formalism are more universal than Hamiltonian or Lagrangian formalism. Finally, the time-scales damped oscillator and a non-Hamiltonian Birkhoffian system are given to exemplify the superiority of the results.

## 1. Introduction

In 1988, Hilger proposed the definition of a time scale $\mathbb{T}$, which is an arbitrary non-empty closed subset of the real numbers $\mathbb{R}$, in order to analyse continuous and discrete systems uniformly [1]. For instance, if we choose a continuous time scale, i.e. $\mathbb{T} = \mathbb{R}$, this time-scale calculus is the same as the calculus of the

classical continuous system; if $\mathbb{T} = \mathbb{Z}$, the calculus is changed to that of the discrete system with step size $\mu = 1$; if $\mathbb{T} = q^{\mathbb{N}_0}$ ($q > 1$), this calculus can solve the problems of quantum systems. Therefore, compared with a single scale, more general results can be obtained based on different time scales. Moreover, the physical essence of those systems can be depicted more accurately by using the time scales theory. The time-scales dynamic equations can provide mathematical models for some processes dependent on continuous-time variables, discrete-time variables and piecewise continuous-time variables, such as the logistic model in biology and the cobweb model in economics [1,2]. Thus it can be seen that the time-scales theory has important theoretical significance and extensive application prospect in various fields [3–6]. Bartosiewicz & Torres [7] found the Noether conserved quantity for delta derivatives based on the time-scales calculus of variation in 2008. It is well known that Noether theorem reveals that conservation quantities of mechanics are directly related to the invariance of actions under infinitesimal transformations. Time-scales Noether theorems are proved not only for delta derivatives, but for nabla derivatives by Martins & Torres [8] in 2010. After that, Malinowska & Martins [9] put forward the second time-scales Noether theorem for delta derivatives in 2013. In the same year, time-scales Noether theorem for delta derivatives of non-conservative non-holonomic systems was studied by Cai *et al.* [10]. Then, the study of time-scales symmetries and conservation quantities was extended to Birkhoffian systems [11] and Hamiltonian systems [12–14].

In the majority of above articles, their variational principles are the classical extremum principles, for example, the famous Hamilton principle, whose action is defined by an integral. However, in general, the Hamilton principle of non-conservative systems is an instability action principle, because the absence of a functional makes its variation equal to zero [15]. Whereas, Herglotz type variational principle can deal with this problem to give the variational description for non-conservative systems by the action functional defined by a differential equation [16]. Based on the Herglotz variational problem, Lagrangians and Hamiltonians with physical meaning can be established for non-conservative systems. Lazo *et al.* obtained the generalized Einstein's field equations for a non-conservative gravity by using the Lagrangian of Herglotz type and applied them to cosmology and gravitational waves [17]. In addition, they constructed Lagrangians of Herglotz type with physical meaning, such as vibrating string under viscous forces, non-conservative electromagnetic theory, non-conservative Schrödinger equation and Klein–Gordon equation, to describe non-conservative systems and quantum systems [18]. Moreover, when these functions do not depend on the action functional, Herglotz variational principle can be reduced to the classical integral variational principle, which can deal with conservative problems. Since Herglotz type variational principle provides a new method for studying non-conservative systems, Herglotz type Noether theorems of mechanical systems have been investigated in recent decades, including non-conservative Lagrangian systems [19,20], non-conservative Hamiltonian systems [21], Birkhoffian systems [15,22], non-conservative non-holonomic systems [23] and other complex systems [24–31]. But so far, time-scales Herglotz variational principle is rarely studied, and the results are limited to Lagrangian formalism [32,33] and Hamiltonian formalism [34].

In 1927, Birkhoff, Poincare's successor, proposed a new type of integral variational principle and a new set of differential equations of motion [35]. The new variational principle was named as Pfaff–Brikhoff principle, and the new equations were called Birkhoff's equations by Santilli [36]. And a mechanical system that describes motion or a physical system that describes state with Birkhoff's equations was called a Birkhoffian system, so Birkhoffian mechanics was born. The new mechanics has a number of nice properties. For example, Birkhoff's equations are not only self-adjoint but also autonomous, and semi-autonomous Birkhoffian systems have Lie algebraic structures and exact symplectic forms [37]. Thus, Birkhoffian mechanics has developed rapidly and has wide applications in many fields, for instance, hadron physics, statistical mechanics, engineering mechanics and biophysics [36]. Another thing to notice is that a Birkhoffian system is a more extensive mechanical system, which can be applicable to the Lagrangian system, Hamiltonian system, holonomic system and non-holonomic system [38].

The pervasiveness of Birkhoffian system motivates us to study time-scales Herglotz variational principle of Birkhoffian systems and its Noether theorem. As we know, all conserved quantities in mechanics are directly related to the invariance of action under a series of infinitesimal transformations, such as energy conservation, momentum conservation and conservation of moment of momentum. The time-scales Herglotz type Noether theorem of Birkhoffian systems has important practical applications for conservative and non-conservative processes in continuous and discrete cases, for example, finding a new solution from a known one, reducing equations, testing computer code and so on. The outline of this paper is as follows. In §2, the time-scales preliminaries of delta derivatives and exponential functions are recalled. Section 3 is our main results: firstly, we introduce the time-scales Herglotz variational problem for delta derivatives of Birkhoffian systems; secondly, the time-scales Herglotz type

Birkhoff's equations for delta derivatives are deduced; then, the time-scales Herglotz type Noether identity and theorem for delta derivatives of Birkhoffian systems are formulated. In §4, the results of Hamiltonian systems and Lagrangian systems are listed to account for the relationship of Hamiltonian, Lagrangian and Birkhoffian systems. Section 5 gives the time-scales damped oscillator of Birkhoffian system and a non-Hamiltonian system as examples. Finally, we offer some conclusions in §6.

## 2. Time-scales preliminaries

A time scale $\mathbb{T}$ is an arbitrary non-empty closed subset of the set $\mathbb{R}$ of real numbers. Let $\mathbb{T}$ be a time scale, for $t \in \mathbb{T}$, the forward jump operator $\sigma : \mathbb{T} \to \mathbb{T}$ is defined by $\sigma(t) = \inf \{s \in \mathbb{T} : s > t\}$ and $\sigma(\sup \mathbb{T}) = \sup \mathbb{T}$, if $\sup \mathbb{T} \in \mathbb{T}$; the backward jump operator $\rho : \mathbb{T} \to \mathbb{T}$ is defined by $\rho(t) = \sup \{s \in \mathbb{T} : s < t\}$ and $\rho(\inf \mathbb{T}) = \inf \mathbb{T}$, if $\inf \mathbb{T} \in \mathbb{T}$. If $\sigma(t) > 0$, $\sigma(t) = 0$, $\rho(t) > 0$, or $\rho(t) = 0$, then $t$ is called right-scattered, right-dense, left-scattered and left-dense, respectively. The graininess function $\mu : \mathbb{T} \to \mathbb{R}$ is defined by $\mu(t) = \sigma(t) - t$, $\mu(t) \geq 0$. For delta derivative, the set $\mathbb{T}^k$ is defined by $\mathbb{T}^k = \mathbb{T} \backslash (\rho(\sup \mathbb{T}), \sup \mathbb{T}]$ if $\sup \mathbb{T} < \infty$, and $\mathbb{T}^k = \mathbb{T}$ if $\sup \mathbb{T} = \infty$. If $f : \mathbb{T} \to \mathbb{R}$ is a function, then $f^\sigma : \mathbb{T} \to \mathbb{R}$ is defined by $f^\sigma(t) = f(\sigma(t))$ for all $t \in \mathbb{T}$, i.e. $f^\sigma = f \circ \sigma$.

**Definition 2.1.** Assume $f : \mathbb{T} \to \mathbb{R}$ is a function and $t \in \mathbb{T}^k$. $f^\Delta(t)$ is called the delta derivative of $f$ at $t$ if for any given $\varepsilon > 0$, there is a neighbourhood $U$ of $t$ (i.e. $U = (t - \delta, t + \delta) \cap \mathbb{T}$) such that

$$|f(\sigma(t)) - f(s) - f^\Delta(t)(\sigma(t) - s)| \leq \varepsilon |\sigma(t) - s| \quad \text{for all } s \in U.$$

Generally, we can denote $f^\Delta(t)$ by $(\Delta/\Delta t)f(t)$. And we call $f$ delta differentiable on $\mathbb{T}^k$ if $f^\Delta(t)$ exists for all $t \in \mathbb{T}^k$. Note that if $\mathbb{T} = \mathbb{R}$, for any $t \in \mathbb{R}$, then $\sigma(t) = \rho(t) = t$, $\mu(t) \equiv 0$ and $f^\Delta(t) = f'(t)$. And if $\mathbb{T} = \mathbb{Z}$, for each $t \in \mathbb{Z}$, then $\sigma(t) = t + 1$, $\mu(t) \equiv 1$ and $f^\Delta(t) = f(t + 1) - f(t)$.

**Definition 2.2.** A function $f : \mathbb{T} \to \mathbb{R}$ is called rd-continuous provided it is continuous at the right-dense points in $\mathbb{T}$ and its left-sided limits exist (finite) at all left-dense points in $\mathbb{T}$. The set of rd-continuous functions $f : \mathbb{T} \to \mathbb{R}$ will be denoted by $C_{rd} = C_{rd}(\mathbb{T}) = C_{rd}(\mathbb{T}, \mathbb{R})$. The set of functions $f : \mathbb{T} \to \mathbb{R}$ that are differentiable and whose derivative is rd-continuous is denoted by $C_{rd}^1 = C_{rd}^1(\mathbb{T}) = C_{rd}^1(\mathbb{T}, \mathbb{R})$.

**Definition 2.3.** A function $f : \mathbb{T} \to \mathbb{R}$ is called regulated provided its right-sided limits exist (finite) at all right-dense points in $\mathbb{T}$ and its left-sided limits exist (finite) at all left-dense points in $\mathbb{T}$. The indefinite integral of a regulated function $f$ is defined by

$$\int f(t)\Delta t = F(t) + c,$$

where $c$ is an arbitrary constant. And the definite integral of $f$ is defined by

$$\int_a^b f(t)\Delta t = F(b) - F(a) \quad \text{for all } a, b \in \mathbb{T}.$$

A function $F : \mathbb{T} \to \mathbb{R}$ is called an antiderivative of $f : \mathbb{T} \to \mathbb{R}$ provided

$$F^\Delta(t) = f(t)$$

holds for all $t \in \mathbb{T}^k$.

**Lemma 2.4.** *Assume $f, g : \mathbb{T} \to \mathbb{R}$ are delta differentiable at $t \in \mathbb{T}^k$. The properties of delta derivatives are:*

$$(f + g)^\Delta(t) = f^\Delta(t) + g^\Delta(t); \tag{2.1}$$

$$(cf)^\Delta(t) = c f^\Delta(t), \quad c \in \mathbb{R}; \tag{2.2}$$

$$(fg)^\Delta(t) = f^\Delta(t)g(t) + f^\sigma(t)g^\Delta(t) = f(t)g^\Delta(t) + f^\Delta(t)g^\sigma(t); \tag{2.3}$$

and

$$\left(\frac{f}{g}\right)^\Delta(t) = \frac{f^\Delta(t)g(t) - f(t)g^\Delta(t)}{g(t)g^\sigma(t)}, \quad g(t)g^\sigma(t) \neq 0. \tag{2.4}$$

**Definition 2.5.** For $h > 0$, $\mathbb{C}_h = \{x \in \mathbb{C} : x \neq -(1/h)\}$, let $\mathbb{Z}_h$ be the strip $\mathbb{Z}_h := \{x \in \mathbb{C} : -(\pi/h) < \mathrm{Im}(x) \leq (\pi/h)\}$. The cylinder transformation $\xi_h : \mathbb{C}_h \to \mathbb{Z}_h$ is defined by

$$\xi_h(x) = \frac{1}{h}\mathrm{Log}(1 + xh).$$

Here, Log is a principal logarithm function. For $h = 0$, let $\mathbb{Z}_0 := \mathbb{C}$, then $\xi_0(x) = x$ is defined for all $x \in \mathbb{C}$.

**Definition 2.6.** A function $\gamma : \mathbb{T} \to \mathbb{R}$ is regressive if $1 + \mu(t)\gamma(t) \neq 0$ for all $t_0 \in \mathbb{T}^k$ holds. If $\gamma \in \mathcal{R}$, the exponential function is defined by

$$e_\gamma(t, s) = \exp\left(\int_s^t \xi_{\mu(\tau)}(\gamma(\theta))\Delta\theta\right) \quad \text{for } s, t \in \mathbb{T}.$$

Here, the set of rd-continuous and regressive functions $f : \mathbb{T} \to \mathbb{R}$ are denoted by $\mathcal{R} = \mathcal{R}(\mathbb{T}) = \mathcal{R}(\mathbb{T}, \mathbb{R})$.

**Lemma 2.7.** If $t, s, r \in \mathbb{T}$, $\gamma \in \mathcal{R}$, and $t_0 \in \mathbb{T}$ is fixed, we list the following properties of exponential functions:

$$e_\gamma^\Delta(t, t_0) = \gamma(t)\, e_\gamma(t, t_0); \tag{2.5}$$

$$e_\gamma^\sigma(t, s) = e_\gamma(\sigma(t), s) = (1 + \mu(t)\gamma(t))\, e_\gamma(t, s); \tag{2.6}$$

$$\frac{1}{e_\gamma(t, s)} = e_\gamma(s, t); \tag{2.7}$$

$$e_\gamma(t, s)\, e_\gamma(s, r) = e_\gamma(t, r); \tag{2.8}$$

and

$$[e_\gamma(s, t)]^\Delta = \frac{-\gamma}{e_\gamma^\sigma(t, s)}. \tag{2.9}$$

**Lemma 2.8.** Suppose $e_\gamma(t, s)$ is regressive. Let $t_1 \in \mathbb{T}$ and $y_1 \in \mathbb{R}$, the unique solution of the initial value problem

$$y^\Delta = \gamma(t)y + f(t) \quad \text{and} \quad y(t_1) = y_1$$

is given by

$$y(t) = e_\gamma(t, t_1)y_1 + \int_{t_1}^t e_\gamma^\sigma(t, \theta) \cdot f(\theta)\Delta\theta.$$

**Lemma 2.9.** Let $g \in C_{\mathrm{rd}}$, $g : [a, b] \to \mathbb{R}^n$, then

$$\int_a^b g^T(t)\eta^\Delta(t)\Delta t = 0 \quad \text{for all } \eta \in C_{\mathrm{rd}}^1 \text{ with } \eta(a) = \eta(b) = 0$$

holds if and only if

$$g(t) = c \quad \text{for } c \in \mathbb{R}^n.$$

The above definitions, lemmas and the specific proof processes of lemmas can be referred to in the literature [1].

# 3. Main results

First, we indicate that the time-scales Herglotz variational problem for delta derivatives of Birkhoffian systems is a functional extremum problem of determining the function $a_\nu(t)$ that extremizes $z(t_2)$, where the action $z(t)$ is a solution of

$$z^\Delta(t) = R_\nu(t, a_\omega^\sigma(t), z(t))a_\nu^\Delta(t) - B(t, a_\omega^\sigma(t), z(t)), \quad (\nu, \omega = 1, 2, \ldots, 2n) \tag{3.1}$$

with the boundary conditions

$$a_\nu(t)|_{t=t_1} = a_{\nu 1}, \quad a_\nu(t)|_{t=t_2} = a_{\nu 2}, \quad (\nu = 1, 2, \ldots, 2n) \tag{3.2}$$

and the initial condition

$$z(t)|_{t=t_1} = z_1. \tag{3.3}$$

Here, $a_\omega^\sigma(t) = (a_\varpi \circ \sigma)(t)$, $t \in \mathbb{T}$, $R_\nu : \mathbb{R} \times \mathbb{R}^{2n} \times \mathbb{R} \to \mathbb{R}$ are the time-scales Herglotz type Birkhoff's functions, and $B : \mathbb{R} \times \mathbb{R}^{2n} \times \mathbb{R} \to \mathbb{R}$ is the time-scales Herglotz type Birkhoffian. $a_{\nu 1}$, $a_{\nu 2}$ and $z_1$ are constants.

**Definition 3.1.** The functional $z$ determined by equation (3.1) is called the time-scales Pfaff–Herglotz action.

Next, we derive the time-scales Herglotz type Birkhoff's equations. From the calculation of isochronous variation on both sides of equation (3.1), it follows that

$$\delta z^\Delta = \left(\frac{\partial R_\nu}{\partial a_\omega^\sigma}\delta a_\omega^\sigma + \frac{\partial R_\nu}{\partial z}\delta z\right)a_\nu^\Delta + R_\nu\delta a_\nu^\Delta - \frac{\partial B}{\partial a_\omega^\sigma}\delta a_\omega^\sigma - \frac{\partial B}{\partial z}\delta z. \tag{3.4}$$

Considering the exchange relationships [10]

$$\frac{\Delta}{\Delta t}(\delta q) = \delta\left(\frac{\Delta}{\Delta t}q\right) = \delta q^\Delta \quad \text{and} \quad (\delta q)^\sigma = \delta q^\sigma.$$

Formula (3.4) can be written as

$$(\delta z)^\Delta = A(t) + \left(\frac{\partial R_\nu}{\partial z}a_\nu^\Delta - \frac{\partial B}{\partial z}\right)\delta z, \tag{3.5}$$

where

$$A(t) = \left(\frac{\partial R_\nu}{\partial a_\omega^\sigma}a_\nu^\Delta - \frac{\partial B}{\partial a_\omega^\sigma}\right)\delta a_\omega^\sigma + R_\nu\delta a_\nu^\Delta. \tag{3.6}$$

According to the condition (3.3), equation (3.5) satisfies the initial value condition

$$\delta z(t_1) = 0. \tag{3.7}$$

Let $\gamma(t) = \frac{\partial R_\nu}{\partial z}a_\nu^\Delta - \frac{\partial B}{\partial z}$, by lemma 2.8 and the properties (2.6), (2.7), (2.8), the solution of equations (3.5) and (3.7) is

$$\delta z(t) = e_\gamma(t, t_1)\int_{t_1}^t e_\gamma^\sigma(t_1, \theta)\cdot A(\theta)\Delta\theta. \tag{3.8}$$

From the boundary conditions (3.2), we have $\delta z(t_2) = 0$. And consider that the action $z(t)$ yields its extremum at $t = t_2$, so that

$$\int_{t_1}^{t_2} e_\gamma^\sigma(t_1, t)\cdot A(t)\Delta t = 0. \tag{3.9}$$

Substituting formula (3.6) into equation (3.9), it follows that

$$\int_{t_1}^{t_2} e_\gamma^\sigma(t_1, t)\left[\left(\frac{\partial R_\nu}{\partial a_\omega^\sigma}a_\nu^\Delta - \frac{\partial B}{\partial a_\omega^\sigma}\right)\delta a_\omega^\sigma + R_\nu\delta a_\nu^\Delta\right]\Delta t = 0. \tag{3.10}$$

From the property (2.3) of delta derivatives, we obtain

$$\int_{t_1}^{t_2} e_\gamma^\sigma(t_1, t)\left(\frac{\partial R_\nu}{\partial a_\omega^\sigma}a_\nu^\Delta - \frac{\partial B}{\partial a_\omega^\sigma}\right)\delta a_\omega^\sigma\Delta t$$

$$= \int_{t_1}^{t_2}\left\{\left[\left(\int_{t_1}^t e_\gamma^\sigma(t_1, \theta)\left(\frac{\partial R_\nu}{\partial a_\omega^\sigma}a_\nu^\Delta - \frac{\partial B}{\partial a_\omega^\sigma}\right)\Delta\theta\right)\delta a_\omega\right]^\Delta\right.$$

$$\left. - \left(\int_{t_1}^t e_\gamma^\sigma(t_1, \theta)\left(\frac{\partial R_\nu}{\partial a_\omega^\sigma}a_\nu^\Delta - \frac{\partial B}{\partial a_\omega^\sigma}\right)\Delta\theta\right)(\delta a_\omega)^\Delta\right\}\Delta t$$

$$= \left[\left(\int_{t_1}^t e_p^\sigma(t_1, \theta)\left(\frac{\partial R_\nu}{\partial a_\omega^\sigma}a_\nu^\Delta - \frac{\partial B}{\partial a_\omega^\sigma}\right)\Delta\theta\right)\delta a_\omega\right]\Bigg|_{t_1}^{t_2}$$

$$- \int_{t_1}^{t_2}\left[\int_{t_1}^t e_p^\sigma(t_1, \theta)\left(\frac{\partial R_\nu}{\partial a_\omega^\sigma}a_\nu^\Delta - \frac{\partial B}{\partial a_\omega^\sigma}\right)\Delta\theta\right](\delta a_\omega)^\Delta\Delta t$$

$$= -\int_{t_1}^{t_2}\left[\int_{t_1}^t e_\gamma^\sigma(t_1, \theta)\left(\frac{\partial R_\nu}{\partial a_\omega^\sigma}a_\nu^\Delta - \frac{\partial B}{\partial a_\omega^\sigma}\right)\Delta\theta\right](\delta a_\omega)^\Delta\Delta t. \tag{3.11}$$

Substituting formula (3.11) into equation (3.10), we have

$$\int_{t_1}^{t_2}\left[e_\gamma^\sigma(t_1, t)R_\omega - \int_{t_1}^t e_\gamma^\sigma(t_1, \theta)\left(\frac{\partial R_\nu}{\partial a_\omega^\sigma}a_\nu^\Delta - \frac{\partial B}{\partial a_\omega^\sigma}\right)\Delta\theta\right](\delta a_\omega)^\Delta\Delta t = 0. \tag{3.12}$$

By lemma 2.9, we obtain

$$e_\gamma^\sigma(t_1, t)R_\omega - \int_{t_1}^t e_\gamma^\sigma(t_1, \theta)\left(\frac{\partial R_\nu}{\partial a_\omega^\sigma}a_\nu^\Delta - \frac{\partial B}{\partial a_\omega^\sigma}\right)\Delta\theta = \text{const},$$

$$(\omega = 1, 2, \ldots, 2n).$$ (3.13)

By delta differentiation of both sides of equation (3.13), we derive the time-scales Herglotz type Birkhoff's equations

$$\frac{\Delta}{\Delta t}\left[e_\gamma^\sigma(t_1, t)R_\omega\right] - e_\gamma^\sigma(t_1, t)\left(\frac{\partial R_\nu}{\partial a_\omega^\sigma}a_\nu^\Delta - \frac{\partial B}{\partial a_\omega^\sigma}\right) = 0, \quad (\omega = 1, 2, \ldots, 2n).$$ (3.14)

**Remark 3.2.** If $\mathbb{T} = \mathbb{R}$, then $\sigma(t) = t$, $\mu(t) = 0$, $e_\gamma^\sigma(t_1, t) = \exp\left(-\int_{t_1}^t ((\partial R_\nu/\partial z)\dot{a}_\nu - (\partial B/\partial z))\mathrm{d}\theta\right)$. Thus, equations (3.14) become the Herglotz type Birkhoff's equations in the continuous case [22]

$$\exp\left[-\int_{t_1}^t \left(\frac{\partial R_\nu}{\partial z}\dot{a}_\nu - \frac{\partial B}{\partial z}\right)\mathrm{d}\theta\right]\left\{\left(\frac{\partial R_\omega}{\partial a_\nu} - \frac{\partial R_\nu}{\partial a_\omega}\right)\dot{a}_\nu + \frac{\partial B}{\partial a_\omega} + \frac{\partial R_\omega}{\partial t}\right.$$
$$\left. + \left(R_\nu\frac{\partial R_\omega}{\partial z} - \frac{\partial R_\nu}{\partial z}R_\omega\right)\dot{a}_\nu - \frac{\partial R_\omega}{\partial z}B + R_\omega\frac{\partial B}{\partial z}\right\} = 0,$$
$$(\omega = 1, 2, \ldots, 2n).$$ (3.15)

**Remark 3.3.** If the time-scales Herglotz type Birkhoffian and Birkhoff's functions do not contain $z$, i.e. $z^\Delta(t) = R_\nu(t, a_\omega^\sigma(t))a_\nu^\Delta(t) - B(t, a_\omega^\sigma(t))$, then $\gamma(t) = 0$, $e_\gamma^\sigma(t_1, t) = 1$. Thus, equations (3.14) change to the time-scales Birkhoff's equations based on the traditional variational problem [11]

$$R_\omega^\Delta - \frac{\partial R_\nu}{\partial a_\omega^\sigma}a_\nu^\Delta + \frac{\partial B}{\partial a_\omega^\sigma} = 0, \quad (\omega = 1, 2, \ldots, 2n).$$ (3.16)

Then, we study the time-scales Herglotz type Noether theorem for delta derivatives of Birkhoffian systems. Let $U$ be a set of $C_{\mathrm{rd}}^1$ functions $a_\nu : [t_1, t_2] \to \mathbb{R}^n$. We introduce the infinitesimal transformations of the one-parameter group with respect to time $t$ on $U$

$$\bar{t} = t + \varepsilon\tau(t, a_\omega, z) \quad \text{and} \quad \bar{a}_\nu(\bar{t}) = a_\nu(t) + \varepsilon\xi_\nu(t, a_\omega, z),$$ (3.17)

where $\tau$, $\xi_\nu$ are infinitesimal generators, and $\varepsilon$ is an infinitesimal parameter. Then, under the transformations (3.17), we can write the time-scales Pfaff–Herglotz action $z(t)$ as $\bar{z}(\bar{t}) = z(t) + \tilde{\Delta}z(t)$, where $\tilde{\Delta}$ denotes total variation. According to the literature [10], we know $\tilde{\Delta}q = \delta q + q^\Delta\tilde{\Delta}t$.

**Definition 3.4.** If the time-scales Pfaff–Herglotz action $z$ is acted on by the infinitesimal transformations (3.17), and $\tilde{\Delta}z(t_b) = 0$ holds for any subinterval $[t_a, t_b] \subseteq [t_1, t_2]$ with $t_a, t_b \in \mathbb{T}$, then the invariance is called the time-scales Herglotz type Noether symmetry of Birkhoffian systems under the infinitesimal transformations.

**Theorem 3.5.** *If the time-scales Pfaff–Herglotz action $z$ is invariant on $U$ under the infinitesimal transformations (3.17), then*

$$\left(\frac{\partial R_\nu}{\partial t}a_\nu^\Delta - \frac{\partial B}{\partial t}\right)\tau + \left(\frac{\partial R_\nu}{\partial a_\omega^\sigma}a_\nu^\Delta - \frac{\partial B}{\partial a_\omega^\sigma}\right)\xi_\omega^\sigma + R_\nu\xi_\nu^\Delta + (\mu R_\nu^\Delta a_\nu^{\Delta\sigma} - B^\sigma)\tau^\Delta = 0$$ (3.18)

*holds for all $t \in [t_1, t_2]$. Formula (3.18) is called the time-scales Herglotz type Noether identity for delta derivatives of Birkhoffian systems.*

*Proof.* On the basis of definition 3.4, we know $\tilde{\Delta}z(t_b) = 0$. Then, from equation (3.1), we obtain

$$\tilde{\Delta}z^\Delta = \tilde{\Delta}(R_\nu a_\nu^\Delta - B) = \tilde{\Delta}R_\nu a_\nu^\Delta + R_\nu\tilde{\Delta}a_\nu^\Delta - \tilde{\Delta}B$$
$$= \left(\frac{\partial R_\nu}{\partial t}\tilde{\Delta}t + \frac{\partial R_\nu}{\partial a_\omega^\sigma}\tilde{\Delta}a_\omega^\sigma + \frac{\partial R_\nu}{\partial z}\tilde{\Delta}z\right)a_\nu^\Delta + R_\nu\tilde{\Delta}a_\nu^\Delta$$
$$- \frac{\partial B}{\partial t}\tilde{\Delta}t - \frac{\partial B}{\partial a_\omega^\sigma}\tilde{\Delta}a_\omega^\sigma - \frac{\partial B}{\partial z}\tilde{\Delta}z.$$ (3.19)

Because of the property (2.3), we can calculate

$$\tilde{\Delta} z^\Delta = \frac{\Delta}{\Delta t}(\tilde{\Delta} z) - z^{\Delta^\sigma} \frac{\Delta}{\Delta t}(\tilde{\Delta} t). \tag{3.20}$$

Taking into account equations (3.20) and (3.1), formula (3.19) can change to

$$\frac{\Delta}{\Delta t}(\tilde{\Delta} z) = \left(\frac{\partial R_\nu}{\partial t}\tilde{\Delta} t + \frac{\partial R_\nu}{\partial a_\omega^\sigma}\tilde{\Delta} a_\omega^\sigma\right)a_\nu^\Delta + R_\nu \tilde{\Delta} a_\nu^\Delta - \frac{\partial B}{\partial t}\tilde{\Delta} t - \frac{\partial B}{\partial a_\omega^\sigma}\tilde{\Delta} a_\omega^\sigma$$
$$+ \left(\frac{\partial R_\nu}{\partial z}a_\nu^\Delta - \frac{\partial B}{\partial z}\right)\tilde{\Delta} z + (R_\nu a_\nu^\Delta - B)^\sigma \frac{\Delta}{\Delta t}(\tilde{\Delta} t), \tag{3.21}$$

where $\tilde{\Delta} t = \varepsilon\tau(t, a_\omega, z)$, $\tilde{\Delta} a_\nu = \varepsilon\xi_\nu(t, a_\omega, z)$. Note the initial condition $\tilde{\Delta} z(t_a) = 0$ that the solution of equation (3.21) is

$$\tilde{\Delta} z(t) = e_\gamma(t, t_a) \int_{t_a}^t e_\gamma^\sigma(t_a, \theta)\left[\left(\frac{\partial R_\nu}{\partial t}\tilde{\Delta} t + \frac{\partial R_\nu}{\partial a_\omega^\sigma}\tilde{\Delta} a_\omega^\sigma\right)a_\nu^\Delta + R_\nu \tilde{\Delta} a_\nu^\Delta\right.$$
$$\left. - \frac{\partial B}{\partial t}\tilde{\Delta} t - \frac{\partial B}{\partial a_\omega^\sigma}\tilde{\Delta} a_\omega^\sigma + (R_\nu a_\nu^\Delta - B)^\sigma \frac{\Delta}{\Delta t}(\tilde{\Delta} t)\right]\Delta\theta. \tag{3.22}$$

Then, when $t = t_b$, we have

$$\int_{t_a}^{t_b} e_\gamma^\sigma(t_a, t)\left[\left(\frac{\partial R_\nu}{\partial t}a_\nu^\Delta - \frac{\partial B}{\partial t}\right)\tau + \left(\frac{\partial R_\nu}{\partial a_\omega^\sigma}a_\nu^\Delta - \frac{\partial B}{\partial a_\omega^\sigma}\right)\xi_\omega^\sigma\right.$$
$$\left. + R_\nu\xi_\nu^\Delta + (\mu R_\nu^\Delta a_\nu^{\Delta\sigma} - B^\sigma)\tau^\Delta\right]\varepsilon\Delta t = 0. \tag{3.23}$$

According to the arbitrariness of integral interval, we obtain

$$e_\gamma^\sigma(t_a, t)\left[\left(\frac{\partial R_\nu}{\partial t}a_\nu^\Delta - \frac{\partial B}{\partial t}\right)\tau + \left(\frac{\partial R_\nu}{\partial a_\omega^\sigma}a_\nu^\Delta - \frac{\partial B}{\partial a_\omega^\sigma}\right)\xi_\omega^\sigma\right.$$
$$\left. + R_\nu\xi_\nu^\Delta + (\mu R_\nu^\Delta a_\nu^{\Delta\sigma} - B^\sigma)\tau^\Delta\right] = 0. \tag{3.24}$$

Since $e_\gamma^\sigma(t_a, t) > 0$, theorem 3.5 is proved. ∎

**Theorem 3.6.** *If the transformations (3.17) correspond to the time-scales Herglotz type Noether symmetry for delta derivatives of Birkhoffian systems, then there exists a conserved quantity in the form of*

$$I_N = e_\gamma^\sigma(t_a, t)R_\omega\xi_\omega + \int_{t_a}^t \left\{\left[e_\gamma^\sigma(t_a, \theta)\left(\frac{\partial R_\nu}{\partial\theta}a_\nu^\Delta - \frac{\partial B}{\partial\theta}\right)\right.\right.$$
$$+ e_\gamma^\sigma(t_a, \theta)\left(\frac{\partial R_\nu}{\partial a_\omega^\sigma}a_\nu^\Delta - \frac{\partial B}{\partial a_\omega^\sigma}\right)a_\omega^\Delta - \frac{\Delta}{\Delta\theta}(e_\gamma^\sigma(t_a, \theta)R_\omega)a_\omega^\Delta\right]\tau$$
$$\left. + e_\gamma^\sigma(t_a, \theta)(\mu R_\nu^\Delta a_\nu^{\Delta\sigma} - B^\sigma)\tau^\Delta\right\}\Delta\theta = \text{const.} \tag{3.25}$$

*Proof.*

$$\frac{\Delta}{\Delta t}I_N = \frac{\Delta}{\Delta t}(e_\gamma^\sigma(t_a, t)R_\omega)\xi_\omega^\sigma + (e_\gamma^\sigma(t_a, t)R_\omega)\xi_\omega^\Delta + \left[e_\gamma^\sigma(t_a, t)\left(\frac{\partial R_\nu}{\partial t}a_\nu^\Delta - \frac{\partial B}{\partial t}\right)\right.$$
$$+ e_\gamma^\sigma(t_a, t)\left(\frac{\partial R_\nu}{\partial a_\omega^\sigma}a_\nu^\Delta - \frac{\partial B}{\partial a_\omega^\sigma}\right)a_\omega^\Delta - \frac{\Delta}{\Delta t}(e_\gamma^\sigma(t_a, t)R_\omega)a_\omega^\Delta\right]\tau + e_\gamma^\sigma(t_a, t)(\mu R_\nu^\Delta a_\nu^{\Delta\sigma} - B^\sigma)\tau^\Delta$$

$$= e_\gamma^\sigma(t_a, t)\left[\left(\frac{\partial R_\nu}{\partial t}a_\nu^\Delta - \frac{\partial B}{\partial t}\right)\tau + R_\nu\xi_\nu^\Delta + (\mu R_\nu^\Delta a_\nu^{\Delta\sigma} - B^\sigma)\tau^\Delta\right]$$
$$+ \frac{\Delta}{\Delta t}(e_\gamma^\sigma(t_a, t)R_\omega)(\xi_\omega^\sigma - a_\omega^\Delta\tau) + e_\gamma^\sigma(t_a, t)\left(\frac{\partial R_\nu}{\partial a_\omega^\sigma}a_\nu^\Delta - \frac{\partial B}{\partial a_\omega^\sigma}\right)a_\omega^\Delta\tau$$

$$= e_\gamma^\sigma(t_a, t)\left[\left(\frac{\partial R_\nu}{\partial t}a_\nu^\Delta - \frac{\partial B}{\partial t}\right)\tau + \left(\frac{\partial R_\nu}{\partial a_\omega^\sigma}a_\nu^\Delta - \frac{\partial B}{\partial a_\omega^\sigma}\right)\xi_\omega^\sigma\right.$$
$$\left. + R_\nu\xi_\nu^\Delta + (\mu R_\nu^\Delta a_\nu^{\Delta\sigma} - B^\sigma)\tau^\Delta\right]$$
$$+ \left[\frac{\Delta}{\Delta t}(e_\gamma^\sigma(t_a, t)R_\omega) - e_\gamma^\sigma(t_a, t)\left(\frac{\partial R_\nu}{\partial a_\omega^\sigma}a_\nu^\Delta - \frac{\partial B}{\partial a_\omega^\sigma}\right)\right](\xi_\omega^\sigma - a_\omega^\Delta\tau).$$

Considering formulae (3.14) and (3.18), we can easily get

$$\frac{\Delta}{\Delta t} I_N = 0.$$

Integrating the above formula, therefore the theorem is proved. ∎

**Remark 3.7.** If $\mathbb{T} = \mathbb{R}$, i.e. $\sigma(t) = t$, $\mu(t) = 0$, then formula (3.18) becomes the Herglotz type Noether identity of Birkhoffian systems in the continuous case [22]

$$\left(\frac{\partial R_\nu}{\partial t} \dot{a}_\nu - \frac{\partial B}{\partial t}\right)\tau + \left(\frac{\partial R_\nu}{\partial a_\omega} \dot{a}_\nu - \frac{\partial B}{\partial a_\omega}\right)\xi_\omega + R_\nu \xi_\nu - B\dot{\tau} = 0. \tag{3.26}$$

And the conserved quantity (3.25) changes to the Herglotz type Noether conserved quantity of classical Birkhoffian systems [22]

$$I_N = \exp\left(-\int_{t_a}^t \left(\frac{\partial R_\nu}{\partial z} \dot{a}_\nu - \frac{\partial B}{\partial z}\right)d\theta\right)(R_\nu \xi_\nu - B\tau) = \text{const.} \tag{3.27}$$

**Remark 3.8.** If $\mathbb{T} = h\mathbb{Z}$, $h > 0$, i.e. $\sigma(t) = t + h$, $\mu(t) = h$, then formula (3.18) can be written as

$$\left(\frac{\partial R_\nu}{\partial t} a_\nu^\Delta - \frac{\partial B}{\partial t}\right)\tau(t) + \left(\frac{\partial R_\nu}{\partial a_\omega^\sigma} a_\nu^\Delta - \frac{\partial B}{\partial a_\omega^\sigma}\right)\xi_\omega(t+h) + R_\nu \xi_\nu^\Delta(t)$$

$$+ [\mu R_\nu^\Delta a_\nu^\Delta(t+h) - B(t+h)]\tau^\Delta(t) = 0. \tag{3.28}$$

And the conserved quantity (3.25) changes to

$$I_N = e_\gamma(t_a, t+h)R_\omega\xi_\omega + \int_{t_a}^t \left\{\left[e_\gamma(t_a, \theta+h)\left(\frac{\partial R_\nu}{\partial \theta} a_\nu^\Delta - \frac{\partial B}{\partial \theta}\right) + e_\gamma(t_a, \theta+h)\right.\right.$$

$$\times \left(\frac{\partial R_\nu}{\partial a_\omega(\theta+h)} a_\nu^\Delta - \frac{\partial B}{\partial a_\omega(\theta+h)}\right)a_\omega^\Delta - \frac{\Delta}{\Delta\theta}(e_\gamma(t_a, \theta+h)R_\omega)a_\omega^\Delta\right]\tau$$

$$+ e_\gamma(t_a, \theta+h)[hR_\nu^\Delta a_\nu^\Delta(\theta+h) - B(\theta+h)]\tau^\Delta\}\Delta\theta = \text{const.} \tag{3.29}$$

Formulae (3.28) and (3.29) are the Herglotz type Noether identity and Noether conserved quantity of Birkhoffian systems in the discrete case.

# 4. Some special cases

The above results of Birkhoffian systems can be applied to Hamiltonian systems and Lagrangian systems under certain cases.

**Case 1:** Let

$$a_\nu^\sigma = \begin{cases} q_\nu^\sigma, & (\nu = 1, 2, \ldots, n) \\ p_{\nu-n}, & (\nu = n+1, n+2, \ldots, 2n), \end{cases} \tag{4.1}$$

$$R_\nu = \begin{cases} p_\nu, & (\nu = 1, 2, \ldots, n) \\ 0, & (\nu = n+1, n+2, \ldots, 2n) \end{cases} \tag{4.2}$$

and $\quad B(t, a_\nu^\sigma(t), z(t)) = H(t, q_s^\sigma(t), p_s(t), z(t)), \quad (s = 1, 2, \ldots, n). \tag{4.3}$

Here, $H$ is the time-scales Hamiltonian for delta derivatives, $q_s (s = 1, 2, \ldots, n)$ are generalized coordinates, and $p_s (s = 1, 2, \ldots, n)$ are generalized momenta. From equations (3.14), we can obtain

$$\frac{\partial H}{\partial p_s} - q_s^\Delta = 0, \quad \frac{\Delta}{\Delta t}(e_\gamma^\sigma(t_1, t) p_s) + e_\gamma^\sigma(t_1, t)\frac{\partial H}{\partial q_s^\sigma} = 0, \quad (s = 1, 2, \ldots, n), \tag{4.4}$$

where $\gamma(t) = -(\partial H/\partial z)$. Equations (4.4) are the time-scales Herglotz type Hamilton canonical equations for delta derivatives [34].

Next, the transformations (3.17) in phase space can be expressed as

$$\left.\begin{aligned} \bar{t} &= t + \varepsilon\tau(t, q_j, p_j, z), \\ \bar{q}_s(\bar{t}) &= q_s(t) + \varepsilon\xi_s(t, q_j, p_j, z) \\ \bar{p}_s(\bar{t}) &= p_s(t) + \varepsilon\eta_s(t, q_j, p_j, z), \end{aligned}\right\} \tag{4.5}$$

and

where $\tau$, $\xi_s$, $\eta_s$ are infinitesimal generators. By theorems 3.5 and 3.6, we can also obtain the time-scales Herglotz type Noether identity and Noether conserved quantity for delta derivatives of Hamiltonian systems.

**Theorem 4.1.** *If the time-scales Hamilton-Herglotz action z of Hamiltonian systems is invariant on U under the infinitesimal transformations (4.5), then*

$$q_s^\Delta \eta_s + p_s \xi_s^\Delta - \frac{\partial H}{\partial t} \tau - \frac{\partial H}{\partial q_s^\sigma} \xi_s^\sigma - \frac{\partial H}{\partial p_s} \eta_s + ((p_s q_s^\Delta)^\sigma - p_s q_s^{\Delta\sigma} - H^\sigma)\tau^\Delta = 0 \qquad (4.6)$$

*holds for all $t \in [t_1, t_2]$. Formula (4.6) is called the time-scales Herglotz type Noether identity for delta derivatives of Hamiltonian systems.*

**Theorem 4.2.** *If the transformations (4.5) correspond to the Herglotz type Noether symmetry for delta derivatives of Hamiltonian systems, then there exists a conserved quantity in the form of*

$$I_N = \int_{t_a}^t \left\{ e_\gamma^\sigma(t_a, \theta) \left[ (p_s q_s^\Delta - H)^\sigma \tau^\Delta - \left( \frac{\partial H}{\partial \theta} + \frac{\partial H}{\partial q_s^\sigma} q_s^\Delta - p_s q_s^{\Delta\Delta} \right) \tau \right] \right.$$
$$\left. + \mu \frac{\Delta}{\Delta\theta} [e_\gamma^\sigma(t_a, \theta) p_s] \cdot \frac{\Delta}{\Delta\theta} (\xi_s - q_s^\Delta \tau) \right\} \Delta\theta + e_\gamma^\sigma(t_a, t) p_s (\xi_s - q_s^\Delta \tau)$$
$$= \text{const.} \qquad (4.7)$$

**Remark 4.3.** *If $\mathbb{T} = \mathbb{R}$, i.e. $\sigma(t) = t$, $\mu(t) = 0$, $e_\gamma^\sigma(t_a, t) = \exp\left(\int_{t_a}^t (\partial H/\partial z)\, dt\right)$, then formula (4.6) becomes the Herglotz type Noether identity of Hamiltonian systems in the continuous case*

$$\dot{q}_s \eta_s + p_s \dot{\xi}_s - \frac{\partial H}{\partial t} \tau - \frac{\partial H}{\partial q_s} \xi_s - \frac{\partial H}{\partial p_s} \eta_s - H\dot{\tau} = 0, \qquad (4.8)$$

*and the conserved quantity (4.7) changes to the Herglotz type Noether conserved quantity of Hamiltonian systems* [21]

$$I_N = \exp\left( \int_{t_a}^t \frac{\partial H}{\partial z} d\theta \right) (p_s \xi_s - H\tau) = \text{const.} \qquad (4.9)$$

**Case 2:** Let

$$H(t, q_s^\sigma(t), p_s(t), z(t)) = p_s q_s^\Delta - L(t, q_s^\sigma(t), q_s^\Delta(t), z(t)) \qquad (4.10)$$

and

$$p_s = \frac{\partial L}{\partial q_s^\Delta}, \quad (s = 1, 2, \ldots, n). \qquad (4.11)$$

The partial derivative of equation (4.10) with respect to $q_s^\sigma$, $p_s$, $z$, respectively, we have

$$\frac{\partial H}{\partial q_s^\sigma} = -\frac{\partial L}{\partial q_s^\sigma}, \quad \frac{\partial H}{\partial p_s} = q_s^\Delta, \quad \frac{\partial H}{\partial z} = -\frac{\partial L}{\partial z}. \qquad (4.12)$$

Thus, equations (3.14) become

$$\frac{\Delta}{\Delta t} \left( e_\gamma^\sigma(t_1, t) \frac{\partial L}{\partial q_s^\Delta} \right) - e_\gamma^\sigma(t_1, t) \frac{\partial L}{\partial q_s^\sigma} = 0, \quad (s = 1, 2, \ldots, n), \qquad (4.13)$$

where $\gamma(t) = \frac{\partial L}{\partial z}$.

The infinitesimal transformations (3.17) in configuration space can be expressed as

$$\bar{t} = t + \varepsilon\tau(t, q_j, z), \quad \bar{q}_s(\bar{t}) = q_s(t) + \varepsilon\xi_s(t, q_j, z), \qquad (4.14)$$

where $\tau$ and $\xi_s$ are infinitesimal generators. Similarly it is possible to obtain the time-scales Herglotz type Noether identity and Noether conserved quantity for delta derivatives of Lagrangian systems according to theorems 3.5 and 3.6.

**Theorem 4.4.** *If the time-scales Hamilton-Herglotz action z of Lagrangian systems is invariant on U under the infinitesimal transformations (4.14), then*

$$\frac{\partial L}{\partial t} \tau + \frac{\partial L}{\partial q_s^\sigma} \xi_s^\sigma + \frac{\partial L}{\partial q_s^\Delta} \xi_s^\Delta + \left( L^\sigma - \frac{\partial L}{\partial q_s^\Delta} q_s^{\Delta\sigma} \right) \tau^\Delta = 0 \qquad (4.15)$$

*holds for all $t \in [t_1, t_2]$. Formula (4.15) is called the time-scales Herglotz type Noether identity for delta derivatives of Lagrangian systems* [32].

**Theorem 4.5.** *If the transformations (4.14) correspond to the time-scales Herglotz type Noether symmetry for delta derivatives of Lagrangian systems, then there exists a conserved quantity in the form of*

$$I_N = \int_{t_a}^{t} \left\{ e_\gamma^\sigma(t_1, \theta) \left[ L^\sigma \tau^\Delta + \left( \frac{\partial L}{\partial t} + \frac{\partial L}{\partial q_s^\sigma} q_s^\Delta + \frac{\partial L}{\partial q_s^\Delta} q_s^{\Delta\Delta} \right) \tau \right] \right.$$
$$\left. + \mu \frac{\Delta}{\Delta\theta} \left[ e_\gamma^\sigma(t_a, \theta) \cdot \frac{\partial L}{\partial q_s^\Delta} \right] \cdot \frac{\Delta}{\Delta\theta}(q_s^\Delta \tau) \right\} \Delta\theta + e_\gamma^\sigma(t_a, t) \cdot \frac{\partial L}{\partial q_s^\Delta} \cdot (\xi_s - q_s^\Delta \tau)$$
$$= \text{const.} \tag{4.16}$$

**Remark 4.6.** If $\mathbb{T} = \mathbb{R}$, i.e. $\sigma(t) = t, \mu(t) = 0$, $e_\gamma^\sigma(t_a, t) = \exp\left(-\int_{t_a}^{t} (\partial L/\partial z)\,dt\right)$, then formula (4.15) becomes the Herglotz type Noether identity of Lagrangian systems in the continuous case

$$\frac{\partial L}{\partial t} \tau + \frac{\partial L}{\partial q_s} \xi_s + \frac{\partial L}{\partial \dot{q}_s} \dot{\xi}_s + \left( L - \frac{\partial L}{\partial \dot{q}_s} \dot{q}_s \right) \dot{\tau} = 0. \tag{4.17}$$

And the conserved quantity (4.16) changes to the Herglotz type Noether conserved quantity of Lagrangian systems

$$I_N = \exp\left(-\int_{t_a}^{t} \frac{\partial L}{\partial z}\,d\theta\right) \left[ \frac{\partial L}{\partial \dot{q}_s}(\xi_s - \dot{q}_s\tau) + L\tau \right] = \text{const.} \tag{4.18}$$

# 5. Examples

**Example 5.1.** Suppose the time-scales Herglotz type Birkhoffian and Birkhoff's functions are, respectively,

$$B = \frac{1}{2} \left[ (a_1^\sigma)^2 + (a_2^\sigma)^2 \right] + \alpha z, \quad R_1 = a_2, \quad R_2 = 0, \tag{5.1}$$

where $z^\Delta = a_2^\sigma a_1^\Delta - (1/2)[(a_1^\sigma)^2 + (a_2^\sigma)^2] - \alpha z$, and $\alpha$ is a constant.

When $\mathbb{T} = \mathbb{R}$, then $\sigma(t) = t$, $\mu(t) = 0$, $e_\gamma^\sigma(t_1, t) = \exp[\alpha(t - t_1)]$. From the Herglotz type Birkhoff's equations (3.15), we have

$$\exp[\alpha(t - t_1)] \cdot (\alpha a_2 + \dot{a}_2 + a_1) = 0 \quad \text{and} \quad -\exp[\alpha(t - t_1)] \cdot (\dot{a}_1 - a_2) = 0, \tag{5.2}$$

i.e.

$$\alpha a_2 + \dot{a}_2 + a_1 = 0 \quad \text{and} \quad \dot{a}_1 - a_2 = 0. \tag{5.3}$$

Let $x = a_1$, $\dot{x} = a_2$; the Birkhoff's equations (5.3) can become the damped oscillator

$$\ddot{x} + \alpha\dot{x} + x = 0. \tag{5.4}$$

And its Noether conserved quantity has been given in the literature [22].

When $\mathbb{T} = h\mathbb{Z}, h > 0$, then $\sigma(t) = t + h, \mu(t) = h, e_\gamma^\sigma(t_1, t) = \exp[\alpha(t + h - t_1)]$. From equations (3.14), we obtain

$$\exp[\alpha(t + h - t_1)] \cdot [\alpha a_2^{\sigma\sigma} + (a_2^\sigma)^\Delta + a_1^\sigma] = 0 \quad \text{and} \quad -\exp[\alpha(t + h - t_1)] \cdot (a_1^\Delta - a_2^\sigma) = 0. \tag{5.5}$$

Let $x = a_1$, $x^\Delta = a_2^\sigma$; then

$$x^{\Delta\Delta} + \alpha x^{\Delta\sigma} + x^\sigma = 0. \tag{5.6}$$

The equation (5.6) can be called the time-scales damped oscillator. Now, we study its Noether conserved quantity on time scale $\mathbb{T} = h\mathbb{Z}$.

From equation (3.18), the Herglotz type Noether identity on this time scale is

$$-a_1^\sigma \xi_1^\sigma + (a_1^\Delta - a_2^\sigma)\xi_2^\sigma + a_2^\sigma \xi_1^\Delta + [h(a_2^\sigma)^\Delta a_1^{\Delta\sigma} - B^\sigma]\tau^\Delta = 0. \tag{5.7}$$

The above equation has a solution

$$\tau = 1, \quad \xi_1 = \exp\left(\int_{t_1}^{t} \frac{a_1^\sigma}{a_1^\Delta - h a_1^\sigma} \Delta\theta\right). \tag{5.8}$$

There is no limit to $\xi_2$. Thus, by theorem 3.6, the Herglotz type conserved quantity of the Birkhoffian system on this time scale is

$$I_N = \exp\left(\int_{t_1}^{t} \frac{\alpha a_1^\sigma}{a_1^\Delta - ha_1^\sigma} \Delta\theta\right) a_2^\sigma = \text{const.} \tag{5.9}$$

**Example 5.2.** Let us study a non-Hamiltonian Birkhoffian system, whose time-scales Herglotz type Birkhoffian and Birkhoff's functions for delta derivatives are, respectively,

$$B = \frac{1}{2}(a_3^\sigma)^2 + \frac{1}{2}(a_4^\sigma)^2 - z, \quad R_1 = a_2^\sigma + a_3^\sigma, \ R_2 = a_4^\sigma, \ R_3 = R_4 = 0, \tag{5.10}$$

where $z$ satisfies the differential equation

$$z^\Delta = (a_2^\sigma + a_3^\sigma)a_1^\Delta + a_4^\sigma a_2^\Delta - \frac{1}{2}(a_3^\sigma)^2 - \frac{1}{2}(a_4^\sigma)^2 + z. \tag{5.11}$$

Now, we study the Herglotz type Noether conserved quantity of the non-Hamiltonian system (5.10) on a time scale of

$$\mathbb{T} = \{2^m : m \in \mathbb{Z}\} \cup \{0\}. \tag{5.12}$$

From the time scale (5.12), it is obvious that $\sigma(t) = 2t$ and $\mu(t) = t$. According to definition 2.6, we have $e_\gamma(t_1, t) = \exp\left(-\int_{t_1}^{t}(1/\theta)\text{Log}(1 + \theta)\Delta\theta\right)$, where $\gamma(t) = (\partial R_\nu/\partial z)a_\nu^\Delta - (\partial B/\partial z) = 1$. Then, $e_\gamma^\sigma(t_1, t) = (1 + t)e_\gamma(t_1, t)$. From equations (3.14), the time-scales Herglotz type Birkhoff's equations of the system can be obtained, as follows:

$$\frac{\Delta}{\Delta t}[e_\gamma^\sigma(t_1, t) \cdot (a_2^\sigma + a_3^\sigma)] = 0, \ \frac{\Delta}{\Delta t}[e_\gamma^\sigma(t_1, t) \cdot a_4^\sigma] - e_\gamma^\sigma(t_1, t)a_1^\Delta = 0,$$

$$- e_\gamma^\sigma(t_1, t) \cdot (a_1^\Delta - a_3^\sigma) = 0 \quad \text{and} \quad - e_\gamma^\sigma(t_1, t) \cdot (a_2^\Delta - a_4^\sigma) = 0. \tag{5.13}$$

From equation (3.18), the Herglotz type Noether identity on this time scale is

$$\{\mu[(a_2^\sigma + a_3^\sigma)^\Delta a_1^{\Delta\sigma} + (a_4^\sigma)^\Delta a_2^{\Delta\sigma}] - B^\sigma\}\tau^\Delta + a_1^\Delta \xi_2^\sigma + a_1^\Delta \xi_3^\sigma$$

$$+ a_2^\Delta \xi_4^\sigma - a_3^\sigma \xi_3^\sigma - a_4^\sigma \xi_4^\sigma + (a_2^\sigma + a_3^\sigma)\xi_1^\Delta + a_4^\sigma \xi_2^\Delta = 0. \tag{5.14}$$

Hence, one solution to the above equation is

$$\tau = 1, \quad \xi_1 = 1, \ \xi_2 = 0. \tag{5.15}$$

There is no limit to $\xi_3$ and $\xi_4$. Therefore, the Herglotz type conserved quantity of the system on the time scale can be obtained by theorem 3.6, as follows

$$I_N = e_\gamma^\sigma(t_1, t) \cdot (a_2^\sigma + a_3^\sigma) = \text{const.} \tag{5.16}$$

# 6. Conclusion

The time-scales Herglotz variational principle for delta derivatives of Birkhoffian systems is introduced and its time-scales Pfaff–Herglotz action is put forward. The time-scales Herglotz type Birkhoff's equations (3.14) are obtained, which can reduce to the Herglotz type Birkhoff's equations (3.15) of continuous systems or the time-scales Birkhoff's equations (3.16) based on the traditional variational problem. The time-scales Herglotz type Noether identity and Noether conserved quantity for delta derivatives of Birkhoffian systems, i.e. theorems 3.5 and 3.6, are new main results. Herglotz type Noether identities and Noether conserved quantities of Birkhoffian systems are given in continuous and discrete cases, respectively. On account of the universality of Birkhoffian systems, theorems 3.5, 3.6 of Birkhoffian systems can become theorems 4.1, 4.2 of Hamiltonian systems or theorems 4.4, 4.5 of Lagrangian systems in special cases. Therefore, the correctness and the generality of the results are verified. Because of the advantage of time-scales Herglotz variational principle, the results of this paper not only are applicable to discrete and continuous Birkhoffian systems but also can be used to solve conservative and non-conservative problems. Moreover, it should not be neglected that the results of this paper provide a theoretical basis for computer programming. Similarly, it is possible to expand to study time-scales Herglotz type Noether theorem for nabla derivatives or mixed derivatives by the method of this paper.

Data accessibility. This article does not contain any additional data.
Authors' contributions. X.T. conceived the study, interpreted the results and drafted the manuscript; Y.Z. coordinated the study, helped draft the manuscript and revised it critically for important intellectual content. Both authors gave final approval for publication.
Competing interests. We declare we have no competing interests.
Funding. This work was supported by the National Natural Science Foundation of China (grant no. 11572212).

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
