## [Reviewer comments · Royal Society Open Science]

Review History

RSOS-191248.R0 (Original submission)

Review form: Reviewer 1

Is the manuscript scientifically sound in its present form?

Yes

Are the interpretations and conclusions justified by the results?

Yes

Is the language acceptable?

Yes

Do you have any ethical concerns with this paper?

No

Have you any concerns about statistical analyses in this paper?

No

Recommendation?

Major revision is needed (please make suggestions in comments)

Comments to the Author(s)

The authors studied the time-scales Herglotz type Noether theorem for delta derivatives of Birkhoffian systems. Some interesting results are presented in this manuscript.

But, I am puzzled by the scientific background, meaning and importance of the time-scales Herglotz type Birkhoffian function and Birkhoff's functions for delta derivatives (i.e., Eqs.(5.1) and (5.2)) in your example.

Review form: Reviewer 2**Is the manuscript scientifically sound in its present form?**

No

Are the interpretations and conclusions justified by the results?

No

Is the language acceptable?

No

Do you have any ethical concerns with this paper?

No

Have you any concerns about statistical analyses in this paper?

No

Recommendation?

Major revision is needed (please make suggestions in comments)

Comments to the Author(s)

In the present work, the authors present a time-scale Herglotz type Noether theorem for Birkhoffian systems. Despite the presented results being new, the importance and motivations are not clear. The text of the work should be improved in order to clarify the relevance of the work.

The potential of real applications for a Noether's like theorem for Herglotz Birkhoffian system should be presented. In special, the authors could add a physically meaningful example. The physical or practical application of the only example presented is not discussed. Furthermore, both the Birkhoffian mechanics and the Herglotz variational problem should be better presented in the work to clarify the reader about the importance of the work. Moreover, despite the time-scale and delta derivatives properties are referenced in the text, some definitions are necessary to improve the presentation of the work since time-scale calculus is a relatively new subject of research.

As examples, the definitions of jump operators, graininess and delta-derivatives. Finally, the text of the paper has several misprints and grammatical errors. For example, the enumeration of the equations are in the form (x.y) but are cited in the text as Eq. (z), e.g., Eq. (10) in the text refers to equation (3.1). Before the work can be considered to publications, the authors should highlight the importance of the presented result and improve the presentation of the work.

Decision letter (RSOS-191248.R0)

02-Sep-2019

Dear Professor Zhang,

The editors assigned to your paper ("Time-scales Herglotz type Noether theorem for delta derivatives of Birkhoffian systems") have now received comments from reviewers. We would like you to revise your paper in accordance with the referee and Associate Editor suggestions which can be found below (not including confidential reports to the Editor). Please note this decision does not guarantee eventual acceptance.

Please submit a copy of your revised paper before 25-Sep-2019. Please note that the revision deadline will expire at 00.00am on this date. If we do not hear from you within this time then it will be assumed that the paper has been withdrawn. In exceptional circumstances, extensions may be possible if agreed with the Editorial Office in advance. We do not allow multiple rounds of revision so we urge you to make every effort to fully address all of the comments at this stage. If deemed necessary by the Editors, your manuscript will be sent back to one or more of the original reviewers for assessment. If the original reviewers are not available, we may invite new reviewers.

- Data accessibility

If you wish to submit your supporting data or code to Dryad (<http://datadryad.org/>), or modify your current submission to dryad, please use the following link:
<http://datadryad.org/submit?journalID=RSOS&manu=RSOS-191248>

- Competing interests

- Authors' contributions

- Acknowledgements

- Funding statement

Kind regards,

Andrew Dunn

on behalf of Professor Takashi Suzuki (Associate Editor) and Mark Chaplain (Subject Editor)
openscience@royalsociety.org

Comments to Author:

Reviewers' Comments to Author:

Reviewer: 1

Comments to the Author(s)

The authors studied the time-scales Herglotz type Noether theorem for delta derivatives of Birkhoffian systems. Some interesting results are presented in this manuscript.

But, I am puzzled by the scientific background, meaning and importance of the time-scales Herglotz type Birkhoffian function and Birkhoff's functions for delta derivatives (i.e., Eqs.(5.1) and (5.2)) in your example.

Reviewer: 2

Comments to the Author(s)

In the present work, the authors present a time-scale Herglotz type Noether theorem for Birkhoffian systems. Despite the presented results being new, the importance and motivations are not clear. The text of the work should be improved in order to clarify the relevance of the work. The potential of real applications for a Noether's like theorem for Herglotz Birkhoffian system should be presented. In special, the authors could add a physically meaningful example. The physical or practical application of the only example presented is not discussed. Furthermore, both the Birkhoffian mechanics and the Herglotz variational problem should be better presented in the work to clarify the reader about the importance of the work. Moreover, despite the time-scale and delta derivatives properties are referenced in the text, some definitions are necessary to improve the presentation of the work since time-scale calculus is a relatively new subject of research. As examples, the definitions of jump operators, graininess and delta-derivatives. Finally, the text of the paper has several misprints and grammatical errors. For example, the enumeration of the equations are in the form (x.y) but are cited in the text as Eq. (z), e.g., Eq. (10) in the text refers to equation (3.1). Before the work can be considered to publications, the authors should highlight the importance of the presented result and improve the presentation of the work.

Author's Response to Decision Letter for (RSOS-191248.R0)

See Appendix A.

Decision letter (RSOS-191248.R1)

23-Oct-2019

Dear Professor Zhang,

I am pleased to inform you that your manuscript entitled "Time-scales Herglotz type Noether theorem for delta derivatives of Birkhoffian systems" is now accepted for publication in Royal Society Open Science.

Best regards,

on behalf of Professor Takashi Suzuki (Associate Editor) and Mark Chaplain (Subject Editor)
openscience@royalsociety.org

Appendix A

Response to Reviewers

Full Title: Time-scales Herglotz type Noether theorem for delta derivatives of Birkhoffian systems

First Author: Xue Tian

Corresponding Author: Yi Zhang

MS Reference Number: RSOS-191248

Dear Prof. Andrew Dunn,

Thank you for your letter and for the reviewers' comments concerning our manuscript entitled "Time-scales Herglotz type Noether theorem for delta derivatives of Birkhoffian systems" (MS Reference Number: RSOS-191248). Those comments are all valuable and very helpful for revising and improving our paper. We have studied comments carefully and have made correction which we hope meet with approval. The main corrections in the paper and the responds to the reviewers' comments are as following:

➤ **Comments of Reviewer #1:**

The authors studied the time-scales Herglotz type Noether theorem for delta derivatives of Birkhoffian systems. Some interesting results are presented in this manuscript.

But, I am puzzled by the scientific background, meaning and importance of the time-scales Herglotz type Birkhoffian function and Birkhoff's functions for delta derivatives (i.e., Eqs.(5.1) and (5.2)) in your example.

Response: *Thanks for peer reviewer's comments. According to the comments of the reviewer, we have added another example (the time-scales damped oscillator) as Example 1, whose Birkhoffian and Birkhoff's functions have scientific background, meaning and importance. The original example as Example 2 is a non-Hamiltonian system, but a Birkhoffian system. The purpose of this example is to illustrate that the time-scales Herglotz type Birkhoffian system can solve the problem that cannot be solved by Hamiltonian theory, so as to highlight the superiority and universality of the Birkhoffian system.*

➤ **Comments of Reviewer #2:**

In the present work, the authors present a time-scale Herglotz type Noether theorem for Birkhoffian systems. Despite the presented results being new, the importance and motivations are not clear. The text of the work should be improved in order to clarify the relevance of the work. The potential of real applications for a Noether's like theorem for Herglotz Birkhoffian system should be presented. In special, the authors could add a physically meaningful example. The physical or practical application of the only example presented is not discussed. Furthermore, both the Birkhoffian mechanics and the Herglotz variational problem should be better presented in the work to clarify the reader about the importance of the work. Moreover, despite the time-scale and delta derivatives properties are referenced in the text, some definitions are necessary to improve the presentation of the work since time-scale calculus is a relatively new subject of research. As examples, the definitions of jump operators, graininess and delta-derivatives. Finally, the text of the paper has several misprints and grammatical errors. For example, the enumeration of the equations are in the form (x.y) but are cited in the text as Eq. (z), e.g., Eq. (10) in the text refers to equation (3.1). Before the work can be considered to publications, the authors should highlight the importance of the presented result and improve the presentation of the work.

Response: *Thanks for peer reviewer's comments. According to the comments of the reviewer, we have checked and revised the manuscript carefully. The details are as follows:*

First of all, in order to present the potential of real applications for a Noether's like theorem for Herglotz Birkhoffian system, we have added some words to the fourth paragraph of the introduction as follows:

“The pervasiveness of Birkhoffian system motivates us to study time-scales Herglotz variational principle of Birkhoffian systems and its Noether theorem. As we know, all conserved quantities in mechanics are directly related to the invariance of action under a series of infinitesimal transformations, such as energy conservation, momentum conservation and conservation of moment of momentum. The time-scales Herglotz type Noether theorem of Birkhoffian systems has important practical applications for conservative and non-conservative processes in continuous and discrete cases, for example, finding a new solution from a known one, reducing equations, testing computer code and so on.”

Secondly, in Section 5, we took the original example as Example 2, and added a physically meaningful example as Example 1, as follows:

“Suppose the time-scales Herglotz type Birkhoffian and Birkhoff's functions are respectively

$$B = \frac{1}{2} \left[(a_1^\sigma)^2 + (a_2^\sigma)^2 \right] + \alpha z, R_1 = a_2, R_2 = 0, \quad (5.1)$$

where $z^\Delta = a_2^\sigma a_1^\Delta - \frac{1}{2} \left[(a_1^\sigma)^2 + (a_2^\sigma)^2 \right] - \alpha z$, and α is a constant.

When $\mathbb{T} = \mathbb{R}$, then $\sigma(t) = t$, $\mu(t) = 0$, $e_\gamma^\sigma(t_1, t) = \exp[\alpha(t - t_1)]$. From the Herglotz type Birkhoff's equations (3.15), we have

$$\exp[\alpha(t - t_1)] \cdot (\alpha a_2 + \dot{a}_2 + a_1) = 0, -\exp[\alpha(t - t_1)] \cdot (\dot{a}_1 - a_2) = 0, \quad (5.2)$$

i.e.,

$$\alpha a_2 + \dot{a}_2 + a_1 = 0, \dot{a}_1 - a_2 = 0. \quad (5.3)$$

Let $x = a_1, \dot{x} = a_2$, then the Birkhoff's equations (5.3) can become the damped oscillator

$$\ddot{x} + \alpha \dot{x} + x = 0. \quad (5.4)$$

And its Noether conserved quantity has been given in literature [22].

When $\mathbb{T} = h\mathbb{Z}$, then $\sigma(t) = t + h$, $\mu(t) = h$, $e_\gamma^\sigma(t_1, t) = \exp[\alpha(t + h - t_1)]$. From Eqs. (3.14), we obtain

$$\exp[\alpha(t + h - t_1)] \cdot \left[\alpha a_2^{\sigma\sigma} + (a_2^\sigma)^\Delta + a_1^\sigma \right] = 0, -\exp[\alpha(t + h - t_1)] \cdot (a_1^\Delta - a_2^\sigma) = 0. \quad (5.5)$$

Let $x = a_1, x^\Delta = a_2^\sigma$, then

$$x^{\Delta\Delta} + \alpha x^{\Delta\sigma} + x^\sigma = 0. \quad (5.6)$$

The equation (5.6) can be called time-scales damped oscillator. Now, we study its Noether conserved quantity on time scale $\mathbb{T} = h\mathbb{Z}$.

From Eq. (3.18), the Herglotz type Noether identity on this time scale is

$$-a_1^\sigma \xi_1^\sigma + (a_1^\Delta - a_2^\sigma) \xi_2^\sigma + a_2^\sigma \xi_1^\Delta + \left[h (a_2^\sigma)^\Delta a_1^{\Delta\sigma} - B^\sigma \right] \tau^\Delta = 0. \quad (5.7)$$

The above equation has a solution is

$$\tau = 1, \xi_1 = \exp \left(\int_{t_1}^t \frac{a_1^\sigma}{a_1^\Delta - h a_1^{\Delta\sigma}} \Delta\theta \right). \quad (5.8)$$

There is no limit to ξ_2 . Thus, by Theorem 3.2, the Herglotz type conserved quantity of the Birkhoffian system on this time scale is

$$I_N = \exp\left(\int_{t_1}^t \frac{\alpha a_1^\sigma}{a_1^\Delta - h a_1^\sigma} \Delta \theta\right) a_2^\sigma = \text{const.} \quad (5.9)''$$

Thirdly, in order to clarify to the reader the importance of the Birkhoffian mechanics and the Herglotz variational problem, we have rewritten the second and third paragraphs of the introduction as follows:

“In the majority of above articles, their variational principles are the classical extremum principles, for example, the famous Hamilton principle, whose action is defined by an integral. However, in general, the Hamilton principle of non-conservative systems is an instability action principle, because the absence of a functional makes its variation equal to zero [15]. Whereas, Herglotz type variational principle can deal with this problem to give the variational description for non-conservative systems by the action functional defined by a differential equation [16]. Based on Herglotz variational problem, Lagrangians and Hamiltonians with physical meaning can be established for non-conservative systems. Lazo et al. obtained the generalized Einstein’s field equations for a non-conservative gravity by using the Lagrangian of Herglotz type and applied them to cosmology and gravitational waves [17]. In addition, they constructed Lagrangians of Herglotz type with physical meaning, such as vibrating string under viscous forces, non-conservative electromagnetic theory, non-conservative Schrödinger equation and Klein-Gordon equation, to describe non-conservative systems and quantum systems [18]. Moreover, when these functions do not depend on the action functional, Herglotz variational principle can be reduced to the classical integral variational principle, which can deal with conservative problems. Since Herglotz type variational principle provides a new method for studying non-conservative systems, Herglotz type Noether theorems of mechanical systems have been investigated in recent decades, including non-conservative Lagrangian systems [19,20], non-conservative Hamiltonian systems [21], Birkhoffian systems [15,22], non-conservative non-holonomic systems [23] and other complex systems [24-31]. But so far, time-scales Herglotz variational principle is rarely studied, and the results are limited to Lagrangian formalism [32,33] and Hamiltonian formalism [34].

In 1927, Birkhoff, Poincare's successor, proposed a new type of integral variational principle and a new set of differential equations of motion [35]. The new variational principle was named as Pfaff-Birkhoff principle, and the new equations were called Birkhoff's equations by Santilli [36]. And a mechanical system that describes motion or a physical system that describes state with Birkhoff's equations was called a Birkhoffian

system, so Birkhoffian mechanics was born. The new mechanics has a number of nice properties. For example, not only Birkhoff's equations are self-adjoint, but also autonomous and semi-autonomous Birkhoffian systems have Lie algebraic structures and exact symplectic forms [37]. Thus, Birkhoffian mechanics has developed rapidly and has wide applications in many fields, for instance, hadron physics, statistical mechanics, engineering mechanics, biophysics [36]. Another thing to notice is that a Birkhoffian system is a more extensive mechanical system, which can be applicable to the Lagrangian system, Hamiltonian system, holonomic system and non-holonomic system [38].”

Fourthly, we have added the time-scales definitions of the forward jump operator σ , backward jump operator ρ , graininess function μ , delta derivative $f^\Delta(t)$ (or $\frac{\Delta}{\Delta t} f(t)$) and the set of rd-continuous functions C_{rd}^1 in Section 2, as follows:

“A time scale \mathbb{T} is an arbitrary nonempty closed subset of the set \mathbb{R} of real numbers. Let \mathbb{T} be a time scale, for $t \in \mathbb{T}$, the forward jump operator $\sigma: \mathbb{T} \rightarrow \mathbb{T}$ is defined by $\sigma(t) = \inf \{s \in \mathbb{T} : s > t\}$ and $\sigma(\sup \mathbb{T}) = \sup \mathbb{T}$, if $\sup \mathbb{T} \in \mathbb{T}$; the backward jump operator $\rho: \mathbb{T} \rightarrow \mathbb{T}$ is defined by $\rho(t) = \sup \{s \in \mathbb{T} : s < t\}$ and $\rho(\inf \mathbb{T}) = \inf \mathbb{T}$, if $\inf \mathbb{T} \in \mathbb{T}$. If $\sigma(t) > 0$, $\sigma(t) = 0$, $\rho(t) > 0$, or $\rho(t) = 0$, then t is called right-scattered, right-dense, left-scattered, and left-dense, respectively. The graininess function $\mu: \mathbb{T} \rightarrow \mathbb{R}$ is defined by $\mu(t) = \sigma(t) - t$, $\mu(t) \geq 0$. For delta derivative, the set \mathbb{T}^k is defined by $\mathbb{T}^k = \mathbb{T} \setminus (\rho(\sup \mathbb{T}), \sup \mathbb{T}]$ if $\sup \mathbb{T} < \infty$, and $\mathbb{T}^k = \mathbb{T}$ if $\sup \mathbb{T} = \infty$. If $f: \mathbb{T} \rightarrow \mathbb{R}$ is a function, then $f^\sigma: \mathbb{T} \rightarrow \mathbb{R}$ is defined by $f^\sigma(t) = f(\sigma(t))$ for all $t \in \mathbb{T}$, i.e., $f^\sigma = f \circ \sigma$.

Definition 2.1. Assume $f: \mathbb{T} \rightarrow \mathbb{R}$ is a function and $t \in \mathbb{T}^k$. $f^\Delta(t)$ is called the delta derivative of f at t if for any given $\varepsilon > 0$, there is a neighborhood U of t (i.e., $U = (t - \delta, t + \delta) \cap \mathbb{T}$) such that

$$|f(\sigma(t)) - f(s) - f^\Delta(t)(\sigma(t) - s)| \leq \varepsilon |\sigma(t) - s| \text{ for all } s \in U.$$

Generally, we can denote $f^\Delta(t)$ by $\frac{\Delta}{\Delta t} f(t)$. And we call f delta differentiable

on \mathbb{T}^k if $f^\Delta(t)$ exists for all $t \in \mathbb{T}^k$. Note that if $\mathbb{T} = \mathbb{R}$, for any $t \in \mathbb{R}$, then $\sigma(t) = \rho(t) = t$, $\mu(t) \equiv 0$ and $f^\Delta(t) = f'(t)$. And if $\mathbb{T} = \mathbb{Z}$, for each $t \in \mathbb{Z}$, then $\sigma(t) = t+1$, $\mu(t) \equiv 1$ and $f^\Delta(t) = f(t+1) - f(t)$.

Definition 2.2. A function $f : \mathbb{T} \rightarrow \mathbb{R}$ is called rd-continuous provided it is continuous at the right-dense points in \mathbb{T} and its left-sided limits exist (finite) at all left-dense points in \mathbb{T} . The set of rd-continuous functions $f : \mathbb{T} \rightarrow \mathbb{R}$ will be denoted by $C_{rd} = C_{rd}(\mathbb{T}) = C_{rd}(\mathbb{T}, \mathbb{R})$. The set of functions $f : \mathbb{T} \rightarrow \mathbb{R}$ that are differentiable and whose derivative is rd-continuous is denoted by $C_{rd}^1 = C_{rd}^1(\mathbb{T}) = C_{rd}^1(\mathbb{T}, \mathbb{R})$.

Definition 2.3. A function $f : \mathbb{T} \rightarrow \mathbb{R}$ is called regulated provided its right-sided limits exist (finite) at all right-dense points in \mathbb{T} and its left-sided limits exist (finite) at all left-dense points in \mathbb{T} . The indefinite integral of a regulated function f is defined by

$$\int f(t)\Delta t = F(t) + c,$$

where c is an arbitrary constant. And the definite integral of f is defined by

$$\int_a^b f(t)\Delta t = F(b) - F(a) \text{ for all } a, b \in \mathbb{T}.$$

A function $F : \mathbb{T} \rightarrow \mathbb{R}$ is called an antiderivative of $f : \mathbb{T} \rightarrow \mathbb{R}$ provided

$$F^\Delta(t) = f(t)$$

holds for all $t \in \mathbb{T}^k$.

Finally, we checked the whole text, corrected some misprints and grammatical errors. And the enumeration of the equations have been changed to the form (x.y). Moreover, we have added the following references:

17. Lazo MJ, Paiva J, Amaral JTS, Frederico JSF. 2017 From an Action principle for action-dependent Lagrangians toward nonconservative gravity: Accelerating universe without dark energy. *Phys. Rev. D* **95** (10), 101501. (doi:10.1103/PhysRevD.95.101501)
18. Lazo MJ, Paiva J, Amaral JTS, Frederico JSF. 2018 An action principle for

action-dependent Lagrangians: toward an action principle to non-conservative systems. *J. Math. Phys.* **59** (3), 032902. (doi:10.1063/1.5019936)

35. Birkhoff BD. 1927 Dynamical systems. Providence RI: AMS College Publ.
36. Santilli RM. 1983 Foundations of theoretical mechanics II. New York: Springer-Verlag.
37. Mei FX, Wu HB. 2012 Historical contributions and inspirations of classical mechanics. *Sci. Tech. Rev.* **30** (11), 61-68. (doi: 10.3981/j.issn.1000-7857.2012.11.009)

Special thanks to you for your good comments.

We tried our best to improve the manuscript and made some changes in the manuscript. These changes will not influence the content and framework of the paper.

We appreciate for Editors and Reviewers' warm work earnestly, and hope that the correction will meet with approval.

Once again, thank you very much for your comments and suggestions.

With best regards,

Xue Tian

School of Science

Nanjing University of Science and Technology

Xuanwu District, Nanjing 210094, P. R. China

E-mail: crystaltianxue@njust.edu.cn

Yi Zhang

Ph.D., Professor

College of Civil Engineering

Suzhou University of Science and Technology

Gaoxin District, Suzhou 215011, P. R. China

E-mail: zhy@mail.usts.edu.cn